# MTPA_Unet: Multi-Scale Transformer-Position Attention Retinal Vessel Segmentation Network Joint Transformer and CNN

**DOI:** 10.3390/s22124592

**Published:** 2022-06-17

**Authors:** Yun Jiang, Jing Liang, Tongtong Cheng, Xin Lin, Yuan Zhang, Jinkun Dong

**Affiliations:** College of Computer Science and Engineering, Northwest Normal University, Lanzhou 730070, China; 2020211957@nwnu.edu.cn (T.C.); 2020222054@nwnu.edu.cn (X.L.); 2020222083@nwnu.edu.cn (Y.Z.); 2020222078@nwnu.edu.cn (J.D.)

**Keywords:** retinal vessel segmentation, convolutional neural network, transformer, attention mechanism

## Abstract

Retinal vessel segmentation is extremely important for risk prediction and treatment of many major diseases. Therefore, accurate segmentation of blood vessel features from retinal images can help assist physicians in diagnosis and treatment. Convolutional neural networks are good at extracting local feature information, but the convolutional block receptive field is limited. Transformer, on the other hand, performs well in modeling long-distance dependencies. Therefore, in this paper, a new network model MTPA_Unet is designed from the perspective of extracting connections between local detailed features and making complements using long-distance dependency information, which is applied to the retinal vessel segmentation task. MTPA_Unet uses multi-resolution image input to enable the network to extract information at different levels. The proposed TPA module not only captures long-distance dependencies, but also focuses on the location information of the vessel pixels to facilitate capillary segmentation. The Transformer is combined with the convolutional neural network in a serial approach, and the original MSA module is replaced by the TPA module to achieve finer segmentation. Finally, the network model is evaluated and analyzed on three recognized retinal image datasets DRIVE, CHASE DB1, and STARE. The evaluation metrics were 0.9718, 0.9762, and 0.9773 for accuracy; 0.8410, 0.8437, and 0.8938 for sensitivity; and 0.8318, 0.8164, and 0.8557 for Dice coefficient. Compared with existing retinal image segmentation methods, the proposed method in this paper achieved better vessel segmentation in all of the publicly available fundus datasets tested performance and results.

## 1. Introduction

Automatic segmentation of the retinal vessels plays an important role in the clinical evaluation and diagnosis of many ocular-related diseases. Since the fundus is the only part of the human body where arterioles and capillaries can be directly and centrally observed with the naked eye, morphological information of these retinal vessels, such as thickness, curvature and density, can reflect the occurrence of disease to some extent [1,2]. Studies have shown that the thickness and curvature of retinal vessels are associated with some extent with hypertension and diabetes mellitus. For example, primary hypertension causes spasms and narrowing of the retinal vessels, thickening of the vessel walls, and in severe cases, exudates, hemorrhages, and cotton wool spots [3]. As can be seen in Figure 1, the fundus of the patient’s eye shows symptoms such as exudates and hemorrhagic spots to varying degrees compared to normal fundus images. The degree of fundus lesions is closely related to the duration of hypertension and its severity. Hypertensive retinopathy will show general arterial stenosis to varying degrees in different disease stages, as shown in Figure 1d, which shows the stenosis of the entire venous tree. Diabetic retinopathy is manifested as retinal hemorrhage, exudation, thinning or even blocking of small blood vessels, leading to retinal anemia and hypoxia, thus promoting the appearance of regenerated blood vessels. The existence of new capillaries is an important sign of the further deterioration of diabetic retinopathy. Simultaneously, diabetic retinopathy and macular mutations are also the main causes of vision loss [4]. Therefore, early detection and diagnosis of these lesions are an important tool to prevent the onset and progression of the disease.

However, in current clinical practice, the manual examination is usually relied upon to obtain information on these retinal fundus lesions. This task is not only time-consuming and laborious but also requires a high level of medical skills from the physician. Therefore, the automatic and accurate segmentation of retinal vessels from retinal fundus images to assist physicians in examination and diagnosis is very important and meaningful work. Many researchers have applied machine learning methods to retinal vessel segmentation tasks, such as using high-pass filtering for vessel enhancement [5] and Gabor wavelet filters to segment retina vessels [6]. Some researchers have applied the EM maximum likelihood estimation algorithm [7] and the GMM expectation-maximization algorithm [5] to the retinal vessel and background pixel classification as well. All of these methods have contributed in retinal vessel segmentation, but further improvements are needed in the accuracy and efficiency of retinal vessel segmentation.

With the rapid development in the field of computer vision, deep learning techniques have played an important function in the field of image processing. Compared with traditional machine learning methods, deep convolutional neural networks [8] have a high capability of extracting effective features of data [9]. Based on the classical UNet [10], FCN [11], and ResNet [12], researchers have proposed many improved convolutional neural network methods. UNet++ [13] uses multiple layers of skip connections to capture features at different levels on the structure of encoder–decoder. Wang et al. [14] proposed dual encoding UNet (DEUNet), which significantly enhanced the ability of the network to segment retinal vessels in an end-to-end and pixel-to-pixel manner. Res-UNet [15] added a weighted attention mechanism to the UNet model to better discriminate between retina vessel and background pixel features.

Although convolutional neural networks (CNN) have a strong feature extraction capability, they still suffer from the problem of a limited convolutional kernel field of receptiive. Therefore, CNN is limited to processing local information but cannot focus on global contextual information. In addition, the difficulty of the retinal vessel segmentation task is how to obtain to perform accurate pixel-level classification rather than image-level classification. To solve the above problems, some researchers have introduced the Transformer [16] framework to computer vision tasks. Vision Transformer (ViT) [17] pioneered the use of a pure Transformer architecture to handle image recognition tasks. Based on the ViT architecture, Deit [18] introduced several training strategies which enable ViT to be trained on ImageNet datasets as well. Pyramid Vision Transformer (PVT) [19], which inherits the advantages of CNN and Transformer, uses a convolution-free backbone to handle computer vision tasks. In addition, more research works are dedicated to combine Transformer with CNN to achieve higher accuracy, such as the work of Chen J. [20], Chen B. [21], Valanarasu [22], and others’ work. Currently, Transformer performs well on medical image processing tasks but usually requires pre-trained networks to make the model perform better, as well as a large amount of data to train the model. Although Transformer is good for acquiring long-distance dependencies in images, it is not good at capturing detailed information about the blood vessels in the fundus of the eye. Just a single-minded pursuit of using Transformer may not be suitable for retinal fundus datasets with small amounts of data. Therefore, this paper takes a look at the convolutional neural network and Transformer mechanisms and their respective focuses. Considering that convolutional neural networks are good at capturing detailed local information, Transformer can complement global information as well as contextual information. In the actual feature extraction and recovery process, the connection between local detail information is more beneficial for extracting features, and the long-distance dependent information plays more of a role of information supplement.

In this paper, we propose an MTPA_Unet(Multi-scale Transformer-Position Attention_Unet) network model for retinal vessel segmentation. It consists of a serial combination of Transformer and CNN. Specifically, we first propose a TPA module to replace the traditional Transformer’s multi-headed attention module. Considering that the Transformer structure is not well adapted to retinal datasets with a small number of samples. Therefore, a lightweight positional attention module is added behind TMSA, which is designed to capture the positional information of retinal vessel pixels more precisely. Secondly, the selection of multi-scale information input makes the network sensitive to different scales to achieve better segmentation. After the Transformer for feature extraction, we feed the extracted information into the encoder of the U-shaped network structure for further fine-grained segmentation of the feature map. A multilayer pooling module is added at the end of downsampling to expand the receptive field. At the same time, the information from each stage of downsampling is fused and provided to the higher levels to compensate for the shallow information. A residual connection is used between the encoder and decoder to reduce noise. Finally, the features are recovered and reconstructed by decoders to enable the network structure to output the segmentation results of retinal vessels. We used three public retinal fundus datasets to evaluate MTPA_Unet, respectively, DRIVE, CHASE DB1, and STARE. Experimental results show that the network achieves better segmentation performance. The main research of the paper is as follows:A TPA module is proposed to replace the MSA structure in the traditional Transformer, which not only considers the relationship between long-distance pixels but also focuses on the acquisition of blood vessel pixel position information. The network model is adapted to the fine segmentation task of retinal blood vessels with a small number of samples.The MTPA_Unet network model is proposed, and the Transformer and convolutional neural network are combined to design and apply it to the retinal blood vessel segmentation task. MTPA_Unet can alleviate the limitations exhibited by CNN in modeling long-term dependencies and achieve higher retinal vessel segmentation accuracy.Perform ablation experiments and comparative experiments on three datasets, DRIVE, CHASE DB1 and STARE, and analyze the results. The results show that the network model proposed in this paper achieves better vessel segmentation performance.

The rest of this paper is organized as follows: Section 2 describes the work related to convolutional neural networks and Transformer. Section 3 describes the MTPA_Unet network model for retinal vessel segmentation. The dataset, implementation details, and evaluation metrics of the experiments are described in Section 4. The ablation experiments and comparison experiments are designed, and the results are analyzed in Section 5. The full text is summarized in Section 6.

## 2. Related Work

### 2.1. Related Work of CNN on Image Segmentation

In recent years, CNN-based methods have achieved great success in the field of semantic segmentation and medical image segmentation due to their efficient feature extraction capability and powerful feature representation. In addition to the classic U-Net [10], FCN [11], ResNet [12], and other methods, the ability of the attention mechanism to efficiently extract feature information has attracted many researchers to do a lot of work on this basis. The classic CBAM [23] network combines channel attention and spatial attention as a classical connection. The CA-Net [9] goes further by combining channel, spatial, and location attention and is an integrated attention convolutional neural network for interpretable medical image segmentation. Non-local blocks [24] have the advantage of capturing long-range dependencies and the disadvantage of increased computational volume.

In addition, it is also an important work to make improvements between the accuracy and cost of the attention mechanism; CC-Net [25] uses a novel cross-attention module to capture contextual information and improve computational efficiency. The APNB network proposed by Zhu et al. [26] introduces a pyramidal sampling module into the nonlocal block, which greatly reduces computation and memory consumption and obtains high semantic segmentation results. The DANet [27] is proposed to use self-adaptive integration of local features and their global dependencies. The output of the two attention modules is summed to further improve the feature representation. Recently, Lian et al. [28] improved the residual network and designed a global and locally enhanced residual U-shaped network for accurate segmentation of retinal blood vessels. Zhu et al. [29] proposed the ACE Net, which applied an extended contraction path segmentation network applied to both cell segmentation and retinal vessel segmentation tasks.

Zhang et al. [30] proposed a novel deep network architecture named Bridge-net, which combines recurrent neural network (RNN) and convolutional neural network (CNN) to effectively utilize the context of retinal vessels. Tan et al. [31] introduced skeletal prior and contrast loss and proposed a new network named SkelCon, which is able to improve the integrity and continuity of thin blood vessels. Arsalan et al. [32] designed a dual-stream fusion network (DSF-Net) and a dual-stream aggregation network (DSA-Net) for the task of semantic segmentation of retinal fundus images. Following this, Arsalan et al. proposed a pooling-free residual segmentation network PLRS-Net [33] with stepped convolution to provide a pooling effect for better retinal vessel segmentation sensitivity. DF-Net [34] proposes a feature fusion module to fuse deep features with vessel responses extracted from Frangi filters. This end-to-end network is not only easy to train, but also has good segmentation performance.

The above methods have contributed to different tasks of retinal blood vessels, innovating the method from different perspectives. However, the segmentation of retinal blood vessels with higher accuracy is the goal that researchers have been pursuing, which is also the starting point of the method in this paper. Compared with these methods using convolutional neural networks, the advantage of this paper is that it pays more attention to the complementary meaning of global information, so Transformer is introduced in the method to assist better segmentation.

### 2.2. Related Work of Transformer in Computer Vision

Transformer [16] first played a role in the field of natural language processing, and Vision Transformer (ViT) [17] first applied a pure Transformer architecture directly to a series of image blocks for classification tasks and achieved excellent results. Since then, Transformer has been increasingly used in the field of computer vision. There are two approaches to the design of network architectures, namely, the pure Transformer architecture approach and the combination of Transformer and convolutional neural networks. In terms of networks with pure Transformer structure, DETR [18] successfully used the Transformer as the main building block in the pipeline to obtain a more flexible and simple target detection framework. ConViT [35] mimicked the local dependency of the convolutional layer by introducing a gated positional self-attentive module. A compelling current work, Swin Transformer [36], uses a hierarchical vision Transformer with shifted windows that can be applied as a general backbone for computer vision for tasks such as image classification, target detection, and semantic segmentation. More designs combine the Transformer with CNN for better results. Transunet [20] is a hybrid coding network based on Transformer and CNN, which designs the Transformer in the downsampling part and focuses more on the acquisition of global information. It is highly competitive for medical segmentation tasks such as multi-organ segmentation and heart segmentation. RTNet [37] proposed a relational Transformer module (RTB) with Transformer as the basic unit and designed a network for diabetic retinopathy segmentation. Heo et al. [38], based on ViT, designed a new pooling-based visual Transformer (PiT) with higher model performance and generalization performance. The Swin Transformer was combined with U-Net to obtain a U-Net-like network structure for medical image segmentation, called Swin-Unet [39]. Gao et al. [40] proposed a multiscale Transformer for medical image segmentation, and the proposed bidirectional attention and global multiscale feature fusion made the model perform well on both 2D and 3D datasets. Since Transformer can compensate for the inherent limitations of convolution, the Transformer structure will be able to continue to be advantageous in the field of computer vision.

## 3. Muti-Transformer-Position Attention_Unet Method

Since CNN has great advantages in extracting local information, it is insufficient for feature extraction of long-distance dependencies. Therefore, in order to balance between long-distance dependencies and short-distance dependencies, this paper uses a combination of traditional CNN and Transformer architectures to achieve high-precision segmentation of retinal fundus vessels. The general structure of our proposed MTPA_Unet network model is discussed as follows.

The input to the network is derived from slices of the original retinal vessel images. Due to the high detail information of fundus vessel endings, feature map inputs of different scales are used to enhance the feature extraction capability of the network. These feature maps were input layer by layer into each stage of the Transformer structure, with each layer having image input sizes of 64 × 64, 32 × 32, 16 × 16, and 8 × 8 pixels. Each stage consists of Patch embedding, position encoding, and TPA modules. Furthermore, the results extracted by the Transformer are passed to the encoder in the corresponding CNN network, i.e., the output of each stage corresponds as the input of the encoder block. After the initial extraction of the image features by the Transformer, the advantages of the Transformer for long-distance dependency acquisition are exploited and the shortcomings of the encoder are compensated. Since short-distance dependencies occupy a more important proportion in retinal vessel segmentation, we perform further fine-grained segmentation of the feature map. The encoder block consists of a feature extraction module and a downsampling module. Here, the output of each layer of the encoder block is fused and passed into the underlying multilayer pooling module together with the encoder block of the last layer in order to make use of the information in the shallow layers and to assist in better segmentation. Residual connections are used between the corresponding encoder and decoder blocks in each layer to reduce noise interference. Finally, the processed feature maps are then fed to the decoder for feature reconstruction and recovery by upsampling operations. The parts are described in detail as follows. The overall structure of the network is shown in Figure 2.

### 3.1. Encoder Block

The encoder block consists of a feature extraction module (FE) and a downsampling module (DS). For the input retinal vessel image, it is first passed through the feature extraction module. The FE module is designed to extract the retinal image vessel features, while the number of channels of the input feature map is adjusted, doubling from 32 channels of the input layer by layer to 256 channels of the fourth layer. The process is as follows: for the input feature maps, a 1 × 1 convolutional layer is first used for dimensionality reduction. The image information is extracted using a 3 × 3 convolution and a 3 × 3 transposed convolution for the reduced-dimensional feature map, respectively, and the extracted information is fused. A 1 × 1 bottleneck layer is subsequently used, and the normalization layer is selected for batch normalization and finally activated by the ReLU function. The feature map is output before it is output with the previous input superimposed. The detailed structure is shown in Figure 3.

Since the pooling layer has the advantage of good feature degradation and feature invariance, we introduce a downsampling module (DS) after the FE module, which allows the model to extract a wider range of features and serves for further feature extraction and downsampling of the image. The DS module consists of an adaptive pooling, three consecutive batch normalization layers, a ReLU activation function, and a Conv layer. The detailed structure is shown in Figure 4.

### 3.2. Transformer-Position Attention Module

The Transformer-Position Attention (TPA) module consists of a modified multi-headed attention TMSA and a position attention module. The Patch Embedding layer and the Positional encoding layer, as necessary structures of the Transformer, are described in detail in the TMSA structure description section. The TMSA and location attention modules are described in detail in turn as follows.

#### 3.2.1. TMSA Structure Description

Patch Embedding layer: The PE layer is used to serialize the input image. Specifically, the input image dimension is H×W×C, and H, W, and C denote the height, width, and number of channels, respectively. Firstly, the input image is divided into N blocks of size $P^{2\times C}$, and then it is reshaped into blocks of dimension $N\times P^{2\times C}$.

Since each stage of the TPA module works on a different size of the input feature map, the PE layer is able to downsample the feature map and gradually expand the channel dimension to achieve a hierarchical feature representation. We use PE before each layer of the TPA module except for the first stage, with the aim of using the PE module to scale the feature map spatial dimension and channel dimension. The spatial dimension is reduced by a factor of 4 and the channel dimension is increased by a factor of 2. This process is implemented using a 3 × 3 convolution with a step size of 2 and a padding of 1. The output of each PE layer can be formalized as Equations (Equation 1) and (Equation 2), where *x* and $x^{\prime }$ denote the feature maps before and after processing, respectively:(1)$$x^{\prime }=BN\left(x\cdot \mathrm{Pr}oj(x)\right)$$
(2)$$PE\left(x\right)=Sigmoid\left(Conv2d\left(x^{\prime }\right)\right)\cdot x^{\prime }$$

Positional encoding layer: to make the positional encoding more flexible, we refer to the setting of positional encoding in [41]. Unlike the traditional positional encoding in ViT [17], we use a deep convolution operation of size 3 × 3 with a padding of 1 to obtain the weights in the pixel direction. The weights are then normalized and scaled by a sigmoid function. The positional encoding process can be expressed as Equation (Equation 3):(3)$$\hat{x}=Sigmoid\left(\mathrm{Deep}Conv2d\left(x\right)\right)\cdot x$$

TMSA structure: The main advantage of the Transformer is that it enables the model to focus on semantic information from the global context and to capture contextual information in both absolute and relative positions. Structurally, the Transformer consists mainly of L-layer Multiheaded Self-Attention (MSA) and Multilayer Perceptron (MLP) blocks. The TMSA used in this paper is similar to the traditional MSA. For the input feature map, a set of projections is first used to obtain Q, $Q\in R^{n\times d_{m}}$. A given Q, K and V can be shared among all attention layers. To reduce the computational effort as well as memory pressure, the input $x\in R^{n\times d_{m}}$ is reshaped into a three-dimensional $\hat{x}\in R^{d_{m}\times h\times w}$, and then the spatial resolution is reduced by convolution operations and normalized using layer normalization. For the newly obtained x^∈Rdm×hhss×wwss, two sets of projections are used to obtain K and V, K,V∈Rdm×hhss×wwss, respectively. For the obtained Q, K and V, the 1 × 1 convolution operation is applied to the transpose of Q and K for simulating the interaction between different heads. A normalization operation is done using softmax to generate the matrix of the contextual attention map. To obtain the set of values weighted by the attention weights, the contextual attention map will be multiplied by V. Finally, after layer normalization of the output, TMSA can be expressed as Equation (Equation 4):(4)$$TMSA=LN\left[Soft\max\left[Conv\left(\frac{Q\cdot K^{T}}{\sqrt{d_{k}}}\right)\right]V\right]$$
where ${\sqrt{d}}_{k}$ is the dimensionality of Q, K, and V. Finally, we linearly project the optimized feature mapping, after Equations (Equation 5) and (Equation 6), and add FFN after TMSA to achieve feature transformation and nonlinearity to obtain the final output of TMSA:(5)$$x_{L}^{1}=TMSA\left(x_{L-1}\right)+x_{L-1}$$
(6)$$Y=x_{L}^{1}+FFN\left(LN\left(x_{L}^{1}\right)\right)$$

#### 3.2.2. Description of Location Attention Structure

Due to the presence of more capillaries in retinal fundus images, more details need to be captured in the information extraction. Therefore, we do not simply use TMSA to process the images but add the location attention module afterward. Under the role of modeling with strong contextual information, the global semantic information description is obtained by establishing the connection between long-distance features of fundus vessel pixels. In turn, a more refined retinal vessel segmentation is achieved. Specifically, for a given feature map F, M, N, V, $\left\{M,N,V\right\}\in R^{C\times W\times H}$ can be obtained after 1 × 1 convolution layers, respectively. For the obtained feature maps M, N, there is a vector $Q_{x}$ at any pixel position *x* in *M*. In order not to increase extra computational effort, when calculating the correlation of a pixel in the whole image with position *x*, the feature vectors that are in the same row and column as position x are first searched in the feature map N and saved in the set $D_{x}\in R^{(H\times W-1)\times C}$. The correlation between the pixel position x and the feature vector associated with it is obtained by the calculation as shown in Equation (Equation 7), and the softmax function is further applied on the multiplication result to generate the attention map $S_{Z}\in R^{(H+W-1)\times H\times W}$:(7)$$S_{z}=soft\max\left(Q_{x}D_{i,x}^{T}\right)$$

The attention map and the set $\psi_{x}\in R^{(H\times W-1)\times C}$ of feature vectors in the same column as x in the feature map V are then multiplied to obtain the new feature map $Y^{\prime }$. Finally, the $Y^{\prime }$ is added to the input feature map F to generate the final output feature map Y. See Equation (Equation 8):(8)$$Y=\sum_{i=0}^{H+W-1}S_{z}\cdot \psi_{i.x}+F$$

By using this feature correlation calculation twice, it is possible to obtain global contextual information about each pixel location. This more comprehensive information extraction enhances the Transformer and introduces little computational effort. The structure of the TPA module is shown in Figure 5.

### 3.3. Loss Function

In order to correct the segmentation error which exists between the segmentation results and the given true value, in this paper, we use the Dice loss function to enhance the retinal vessel segmentation results. The Dice coefficient is an ensemble similarity measure function which takes values in the range [0, 1]. It is used in this paper to calculate the difference between the predicted retinal vessel segmentation result (denoted as *P*) and the true value (denoted as *G*), and the Dice coefficient formula is defined as Equation (Equation 9):(9)$$DiceCoefficient=2\times \frac{\left|P\cap G\right|}{\left|P\right|+\left|G\right|}$$
where $\left|P\cap G\right|$ denotes the intersection of the predicted retinal vessel segmentation result and the true value, and |*P*| and |*G*| denote their pixel counts, respectively. The Dice loss function is then deduced from the Dice coefficient, denoted as DiceLoss = 1DiceCoefficient, which is defined as in Equation (Equation 10). A constant w is introduced in the concrete implementation to prevent the denominator from being zero. Because the real goal in the semantic segmentation task is to maximize the Dice Coefficient, in order to improve the segmentation accuracy, it is to minimize the DiceLoss. In addition, since DiceLoss is a region-related loss, that is, the loss of the current pixel is also related to the values of other points. It can also be seen from the definition form of DiceLoss that the loss calculated by the fixed-size positive sample area is the same, and the supervision effect on the network will not change with the size of the image. Therefore, in the training process, DiceLoss is more inclined to mine the foreground area, and the effect may be better for the class imbalance problem:(10)$$DiceLoss=1-2\times \frac{\left|P\cap G\right|+w}{\left|P\right|+\left|G\right|+w}$$

## 4. Dataset and Evaluation Criteria

### 4.1. Dataset

The retinal images used in this paper are from three publicly available datasets, respectively, the DRIVE, CHASEDB1, and STARE dataset. The DRIVE dataset [42] consists of 40 color images of retinal fundus vessels, of which seven images suffer from different degrees of lesions. It also contains groundtruth images and corresponding mask images that were manually segmented by two experts. The size of each image is 565 × 584, and the first twenty fundus images are set as the test set. The last twenty images are set as the training set. The experimental comparison labels were chosen from the manual segmentation results of the first expert.

The STARE dataset [43] consists of 20 color images of retinal fundus vessels, 10 of which suffer from different degrees of lesions. It also contains the groundtruth images manually segmented by two experts and consists of the corresponding mask images. The size of each image was 700 × 605 pixels. The experimental comparison labels were selected from the manual segmentation results of the first expert.

The CHASE DB1 dataset [44] consists of 28 color images of the retinal fundus vessels, with images acquired from the left and right eyes of 14 affected children. It also contains the groundtruth images manually segmented by two experts and consists of the corresponding mask images. The image size was 999 × 960. Twenty images were used as the training set, and the remaining eight images were used as the test set. The experimental comparison labels are chosen from the manual segmentation results of the first expert.

Figure 6 shows three example dataset images, from top to bottom, the CHASE DB1, DRIVE, and STARE dataset, respectively. From left to right are the original retinal fundus vessel medical image, the masked image, and the true value of the expert’s manual segmentation, respectively.

### 4.2. Image Preprocessing

In this paper, we also use the necessary preprocessing to enhance the vessel contours in the original retinal images. In this paper, we used the preprocessing methods proposed by Jiang et al. [45], which are data normalization, adaptive histogram equalization (CLAHE) processing, and gamma correction methods, respectively. It was experimentally verified that the blood vessels in the grayscale images were clearest after fusing the G, R, and B channels in the ratio of 29.9%, 58.7%, and 11.4%. Normalization was used to improve the convergence speed of the model, and CLAHE processing was used to enhance the contrast between the blood vessels and the background in the original images. Finally, gamma correction is used to improve the quality of retinal fundus vessel images. The images processed by the four strategies are shown in Figure 7b–e. Obviously, the blood vessels in the retinal images are clearer, and the contrast with the background is more obvious after the above preprocessing operations.

### 4.3. Experimental Evaluation Metrics

To quantitatively evaluate the accuracy of the method in this paper for the retinal vessel segmentation task, the performance of the evaluation metrics such as Dice coefficient, Accuracy, Sensitivity, and Specificity were analyzed using a confusion matrix. The corresponding equations for each evaluation metric are expressed in Equations (Equation 11)–(Equation 15). In image segmentation tasks, the Dice coefficient is usually used to express the proportional relationship between sensitivity and accuracy, and its value is closer to 1.0 to indicate better segmentation. Accuracy indicates the ratio of the sum of correctly segmented vessel pixels and background pixels to the total pixels of the whole image. The sensitivity indicates the ratio of correctly segmented vessel pixels to the total real vessel pixels, and its value is Specificity, indicating the proportion of correctly segmented background pixels to the total real background pixels, and the value is closer to 1.0, which means the fewer pixels are incorrectly segmented:(11)$$Dice=\frac{2\times TP}{2\times TP+FN+FP}$$
(12)$$Accuracy=\frac{TP+TN}{TP+FN+FP+TN}$$
(13)$$Sensitivity=\frac{TP}{TP+FN}$$
(14)$$Specificity=\frac{TN}{TN+FP}$$
(15)$$Precision=\frac{TP}{TP+FP}$$
where true positive (*TP*) is the number of vessel pixels that are correctly segmented, true negative (*TN*) is the number of background pixels that are correctly segmented, false positive (*FP*) is the number of background pixels that are incorrectly segmented as vessel pixels, and false negative (*FN*) is the number of vessel pixels that are incorrectly segmented as background pixels.

## 5. Experimental Results and Analysis

### 5.1. Experimental Environment and Parameter Settings

The method in this paper is implemented using the Pytorch framework for deep learning. The model training was implemented on a Quadro RTX 6000 server with a GPU memory size of 24 GB and an operating system of Ubuntu64. The initial learning rate of 0.001 was used for training. We used the model with the best validation performance in the test, and the Dice loss function was used for the loss function.

For the DRIVE and CHASE DB1 datasets, the number of iterations of the model is set to 100, the training batch size is 32, and the threshold is set to 0.5. Since there are only 20 images in the STARE dataset, the experiments are conducted using the leave-one-out method to make the training effect as good as possible. That is, one image is used for training at a time, and the remaining 19 samples are used for testing. The training batch size was set to 64, the number of iterations of the model was set to 100, and the threshold value was set to 0.48.

### 5.2. Experimental Comparison of Ablation Structures

In order to verify the effectiveness of the utilization of the shallow information in the model MTPA_Unet, the TPA module, and the Transformer for the retinal vessel segmentation task in this paper, in the same experimental setting, we performed retinal vessel segmentation experiments in DRIVE, CHASE DB1, and STARE datasets, respectively, using the U-network as the baseline model. The performance of these modules was quantified by designing ablation experiments. First, the baseline network is based on a modification of the U-Net incorporating residual connectivity and a multiscale pooling module, denoted as BaseLine. Based on this, the output of each encoder block is fused and passed to the multiscale pooling module to utilize shallow coarse-grained feature information. Next, the Transformer is added to the BaseLine+SCI to compensate for the contextual information, and a multi-scale network input is used. Finally, the MSA is replaced by the TPA structure, which is the model MTPA_Unet in this paper. MTPA_Unet (w/o pre) means operating directly on the original image without any preprocessing.

The structure of the ablation experiments performed on the DRIVE and CHASE DB1 datasets are presented in Table 1 and Table 2, respectively. The bolded data in the tables indicate the maximum values achieved by the different network models on the corresponding evaluation metrics. As far as the performance of BaseLine is concerned, the Dice coefficient and sensitivity reached 0.8136 and 0.8266, 0.8324, and 0.8060 on the DRIVE and CHASE DB1 datasets, respectively. Model BaseLine+SCI reached 0.8289 and 0.8278 on the DRIVE dataset for Dice coefficient and sensitivity, respectively. The sensitivity increased by 1.18%. On the CHASE DB1 dataset, the Dice coefficient and sensitivity reached 0.8143 and 0.8296, respectively, with some decrease in sensitivity but a small increase in Dice coefficient, while the rest of the metrics were basically the same. This proves that the inclusion of shallow information is beneficial to segment the vessel pixels from the background pixels and can capture more details of the vessels. In order to combine the long-distance dependency extraction capability of Transformer with the local information extraction capability of CNN, we added a Transformer structure based on the BaseLine+SCI model and used multi-scale input in order to exploit the multi-resolution feature of retinal vessel images. It can be seen that the Dice coefficient and sensitivity reach 0.8300 and 0.8249 on the DRIVE dataset, and 0.8139 and 0.8434 on the CHASE DB1 dataset, respectively. This proves that Transformer is helpful for the vessel segmentation task. However, retinal vessel segmentation has more vessel branches as well as terminal parts compared with other medical image segmentation tasks. The acquisition of vessel location information is insufficient using only the Transformer structure. Therefore, after adding the location attention, the Dice coefficient and sensitivity of model MTPA_Unet on DRIVE, CHASE DB1 datasets reach 0.8318 and 0.8164, 0.8410 and 0.8437, respectively, which are improved by 0.18% and 0.21%, 1.65% and 0.04%, respectively. It is further demonstrated that our proposed module is effective in extracting the contextual information of retinal images. Multiple retinal vessel section images at different scales also enable the model to learn different characteristics as well as fine vessel features.

For the STARE dataset, the same ablation experiments were designed. For clarity, the test results of the MTP_Unet model on 20 images are listed in Table 3. The results of the 20 tests on the five metrics are averaged as the test results of MTPA_Unet on the STARE dataset.

For the models BaseLine, BaseLine+SCI, and BaseLine+SCI+MT were trained and tested on the STARE dataset using the leave-one-out method. For the sake of simplicity, only the average values obtained on the five evaluated metrics are shown here. The experimental results are shown in Table 4, with the highest value for each metric in bold. Comparing the performance of BaseLine, Dice and sensitivity improved by 0.46% and 0.71%, respectively, after adding the shallow information. Combining the multiscale Transformer with it increases Dice and sensitivity by 0.09% and 0.3%, respectively. The method MTPA_Unet in this paper improves Dice and sensitivity by 1.23% and 1.26%, respectively, on this basis. The combined performance shows that the method in this paper can improve in each index and is very effective for more accurate segmentation of vessel pixels.

In addition to using these evaluation metrics to measure the effectiveness of the models, to see more clearly the segmentation effect of the retinal fundus vessels, the segmentation effect of each model on the same data set is shown in Figure 8. The visualization of the segmentation results shows details that are not reflected in the numerical data. Columns (a)–(f) in Figure 8 show the original retinal vessel image, the true value of manual segmentation by professionals, the segmentation result of BaseLine, and the segmentation result of BaseLine+SCI, the segmentation result of BaseLine+SCI+ MT, and the segmentation result of MTPA_Unet, respectively. From top to bottom, the segmentation results of medical images on CHASE DB1, DRIVE, and STARE datasets are shown in order. To highlight the model segmentation effect, the capillary segmentation region is highlighted in the visualization comparison. Some regions in the original retinal image, the baseline image, and each model segmentation result map are shown enlarged and marked with red boxes.

It can be seen from the performance of each model on the three datasets in Figure 8. The difficulty of retinal fundus vessel segmentation is the accurate segmentation of the surrounding fine vessel branches and some interlacing locations of vessels. In contrast, segmentation of the thicker veins or slightly thinner arteries in the center of the retina is easier for the common model. The addition of the Transformer structure brings more information to the network and enables more detailed segmentation of the blood vessels, which is not possible with the baseline model. The model in this paper obtains better segmentation results compared with the above ablation structures. This fully demonstrates that considering the information of vessel pixel location can enhance the information extraction ability of Transformer structure and make the network have better segmentation ability. It is clearly observed from the comparison of the regions marked with red boxes in the figure that the segmentation of the border part and capillary part of the blood vessels is more accurate and clear. The above facts show that the network structure proposed in this paper is feasible and effective in a real segmentation task. The improved network model is able to obtain better segmentation results on all three datasets.

### 5.3. Comparison with Existing Models

To further illustrate the model validity, the method in this paper conducts comparison experiments with some existing state-of-the-art methods on three datasets. Classical advanced methods such as U-Net++ [13], R2U-Net [46], CA-Net [9], and SCS-Net [47] within the last five years were selected. The evaluation was carried out according to four evaluation metrics, namely Dice, Accuracy, Sensitivity, and Specificity. Table 5, Table 6 and Table 7 show the evaluation results of different models on the DRIVE, STARE, and CHASE DB1 datasets, respectively, for the retinal vessel segmentation task.

As can be seen from Table 5, on the DRIVE dataset, the method MTPA_Unet in this paper performs the best on the three metrics of accuracy, sensitivity, and Dice. Compared with the suboptimal method, the improvement is 0.08%, 1.21%, and 0.16%, respectively. As can be seen in Table 7, on the CHASE DB1 dataset, the method in this paper performs better in terms of accuracy and Dice. The improvement is 0.02% and 0.25%, respectively. From Table 6, it can be seen that, on the STARE dataset, the method in this paper performs best on two metrics, Sensitivity and Dice. Compared with the suboptimal method, it improves 2.52% in sensitivity and 0.82% in Dice. The improvement of the evaluation index results verifies the effectiveness of the method in this paper. The superior sensitivity metric results prove the better accuracy of the method in this paper for the correct classification of vessel pixels.

Similarly, the segmentation capability of the model is visualized with visualized images of retinal vessel segmentation results. The method in this paper is compared with the current better performing UNet++ [13], CA-Net [9], and AG-UNet [52] network models for the segmentation task of retinal fundus vessels for visualization. Figure 9 and Figure 10 show the comparison of the visualization results of different models on the DRIVE and CHASE DB1 datasets for the retinal vessel segmentation task, respectively. Columns (a)–(f) show the segmentation results of the original retinal vessel images, the true values manually segmented by a professional, UNet++, CA-Net, AG-UNet, and MTPA_Unet, respectively. On the DRIVE and CHASE DB1 datasets, we can see that CA-Net and AG-UNet can basically segment all arteries and veins, but there are still some blood vessels that are not segmented, and the background pixels are incorrectly segmented as blood vessel pixels, and the noise is more obvious in the results of CA-Net segmentation, while UNet++ performs well in the evaluation metrics and visualization segmentation results. However, the segmentation of some of the vascular details is slightly inferior. In contrast, MTPA_Unet performs better in the segmentation of small blood vessels because it fully utilizes the inter-pixel position relationship and multi-scale feature maps to reduce the pixel misclassification problem. Since MTPA_Unet takes into account the information of deep and shallow layers, the noise effect is reduced in the segmentation results.

Figure 11 shows the comparison of the visualization results of different models on the STARE dataset for the retinal vessel segmentation task. Columns (a)–(e) show the original retinal vessel images, the true values manually segmented by professionals, the segmentation results of UNet++, CA-Net, and the segmentation results of MTPA_Unet, respectively. The method in this paper can obtain clearer vessel segmentation results compared with UNet++ and CA-Net. The segmentation of capillaries is more accurate. The segmentation is more smooth in the articulation part of some vessels. This is due to the fact that this method takes into account the long-range relationship between pixels and the local relationship, and focuses on the location connected with the surrounding pixels when obtaining the pixel information of blood vessels. The comparative analysis shows that the method in this paper has better performance and advantages for the retinal vessel segmentation task. This conclusion can be clearly derived from Figure 9 and Figure 11.

### 5.4. Analysis of the Number of Model Parameters and Evaluation of ROC Curves

We evaluate the cost of the network model to obtain better segmentation performance from the perspective of the model parameters. Ordinary CNN networks usually do not introduce too much computation, while Transformer leads to a higher number of parameters due to the complex multi-headed attention computation. To demonstrate that our final model MTPA_Unet does not introduce too many parameters, the obtained high precision experimental results do not only rely on the complex model to be achieved. The model with the introduction of the Transformer structure is compared with the number of MTPA_Unet parameters with the addition of the position attention module. As can be seen in Table 8, the Transformer network model with the addition of multiscale inputs brings a higher number of parameters compared to the CNN network. However, the MTPA_Unet modification of the Transformer not only does not introduce too many parameters but also enables the model to achieve higher accuracy in the retinal vessel segmentation task. This proves that the method in this paper is not complicated and effective.

To further judge the model performance, Receiver Operating Characteristic (ROC) curves and Precision Recall (PR) curves were calculated for each ablation structure network model and displayed in visualization in Figure 12. The ROC curves express the information between the incorrect segmentation of background pixels into vascular pixels and the correct segmentation of vascular pixels. When the proportion of these two is larger, the PR curve can better reflect the real situation of pixel classification. As far as the experimental results are concerned, the area occupied by the ROC curve and PR curve of MTPA_Unet is the largest in all three data sets. This indicates that the method in this paper achieves the best results on the retinal vessel segmentation task, and is able to utilize the long-distance dependence and local information in combination. It is also able to extract the positional relationships between retinal vessel pixels and take into account the deep and shallow feature information, resulting in the best performance of the model.

## 6. Conclusions

The MTPA_Unet retinal vessel segmentation network model proposed in this paper jointly uses Transformer and convolutional neural network to help improve the performance of the network model. Since the connection between two distant pixels on an image is important for more accurate retinal vessel segmentation, the convolutional neural network is utilized to extract the long-distance dependencies while taking advantage of the convolutional neural network for local information extraction. The proposed TPA module can further enhance the acquisition of retinal vessel location information, having richer feature information to be fully used in the refinement process. The multi-resolution image input and the utilization of shallow feature information further alleviate the problems of blurred boundaries of segmentation results and inaccurate capillary segmentation. We trained and tested the MTPA_Unet network model proposed in this paper on the DRIVE, CHASEDB1, and STARE datasets. The evaluation shows that the model has achieved good results in terms of Accuracy and Dice. Comparison experiments were also designed to compare and analyze the evaluation results with other popular methods to visually demonstrate the segmentation details of each network model on the retinal vessel task. The comparison of the segmentation results and the analytical discussion show that the MTPA_Unet network model proposed in this paper is more advantageous compared with other methods. Future research will aim to further improve the accuracy of the network model for the retinal vessel segmentation task without sacrificing time and storage.

## Figures and Tables

**Figure 1 sensors-22-04592-f001:**
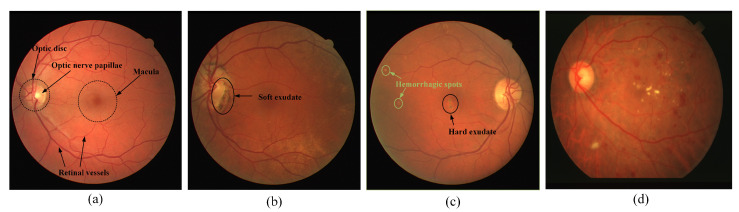
Retinal fundus images. (**a**) normal fundus image; (**b**) background diabetic retinopathy, pigment epithelial atrophy; (**c**) choroidal lesion; (**d**) narrowing, entire venous tree.

**Figure 2 sensors-22-04592-f002:**
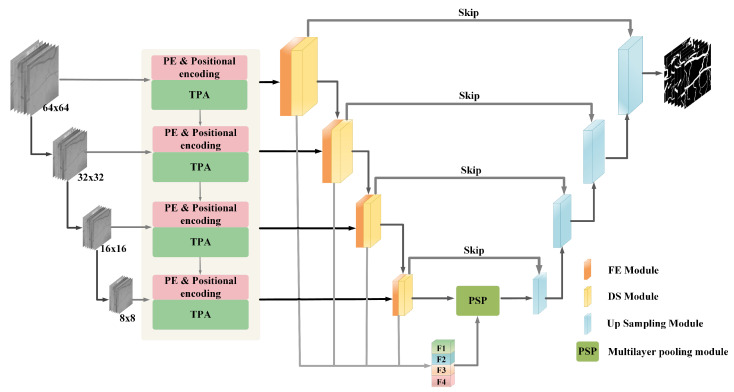
Figure of downsampling module.

**Figure 3 sensors-22-04592-f003:**
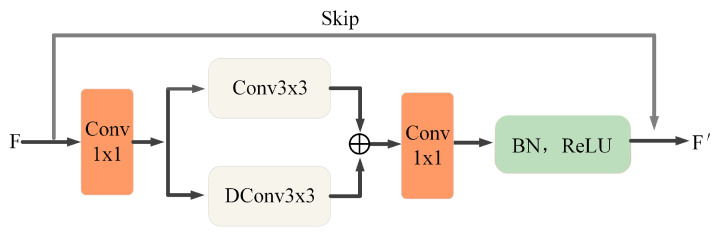
Feature extraction module.

**Figure 4 sensors-22-04592-f004:**
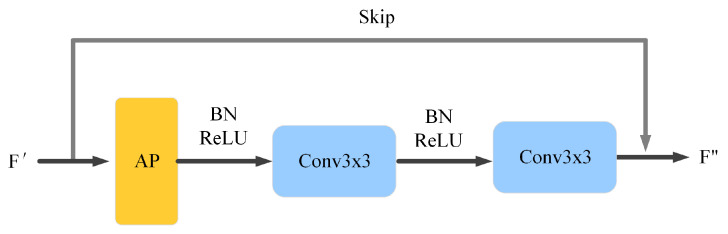
Downsampling module.

**Figure 5 sensors-22-04592-f005:**
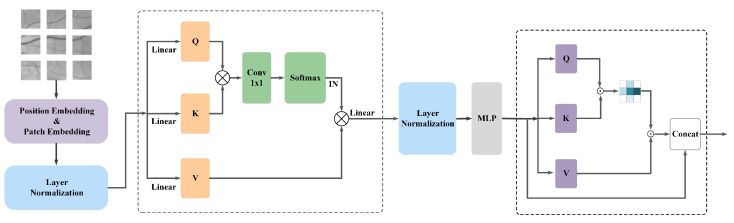
Structure of TPA module.

**Figure 6 sensors-22-04592-f006:**
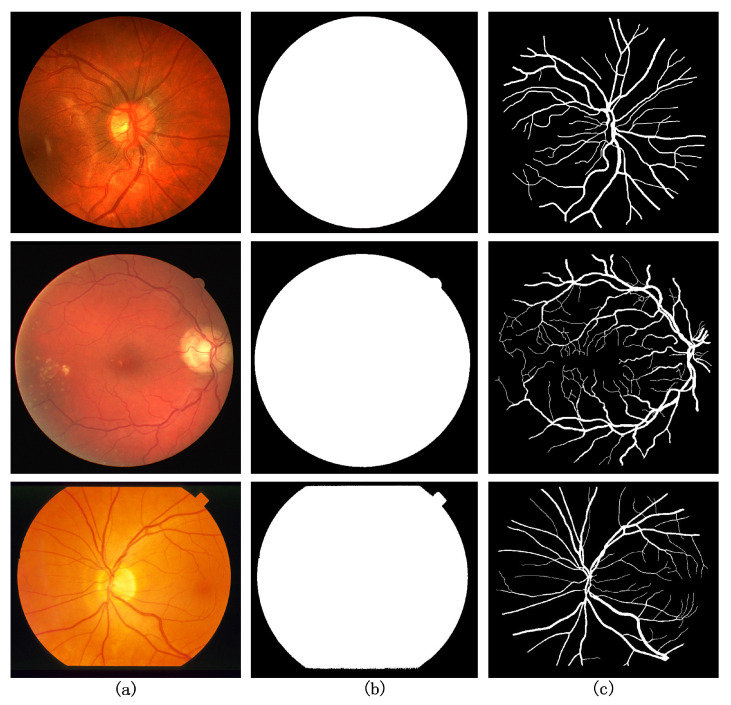
Example images of three datasets (**a**) original retinal fundus vessel medical image, (**b**) masked image, and (**c**) expert manual segmentation of the groundtruth.

**Figure 7 sensors-22-04592-f007:**
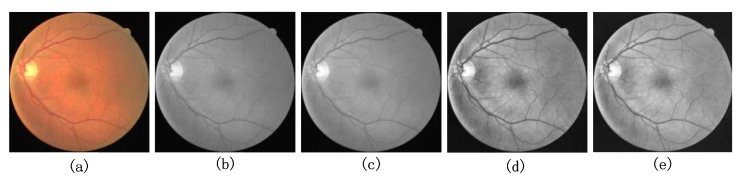
Pre-processing results of (**a**) original retinal fundus vessel medical image, (**b**) RGB three-channel scaled fusion image, (**c**) data normalized image, (**d**) CLAHE processed image, and (**e**) gamma corrected image.

**Figure 8 sensors-22-04592-f008:**
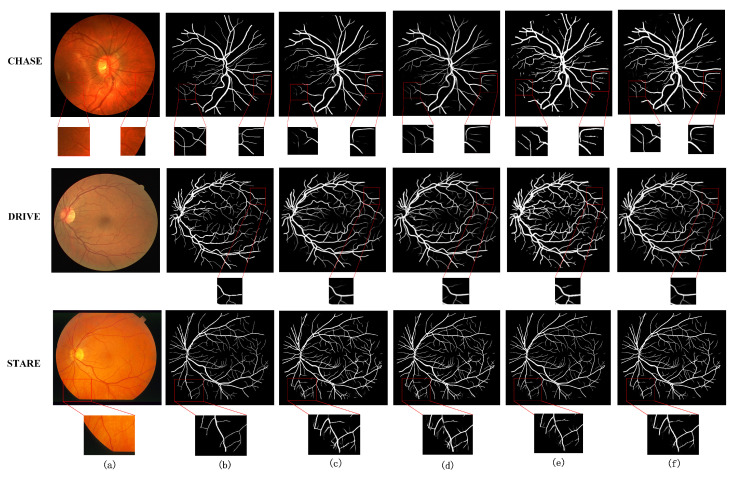
Visualization of ablation experiments on three datasets. (**a**) original image, (**b**) groundtruth image, (**c**) BaseLine, (**d**) BaseLine+SCI, (**e**) BaseLine+SCI+ MT, (**f**) MTPA_Unet.

**Figure 9 sensors-22-04592-f009:**
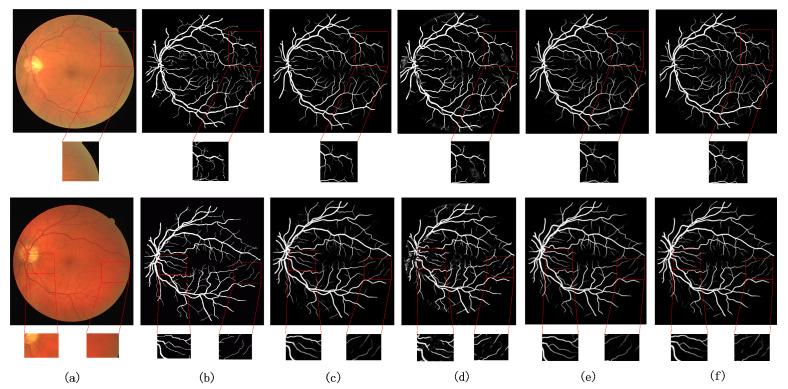
Comparison of visualization results with other methods on DRIVE dataset (**a**) original image, (**b**) groundtruth image, (**c**) UNet++ [13], (**d**) CA-Net [9], (**e**) AG-UNet [52], (**f**) Ours.

**Figure 10 sensors-22-04592-f010:**
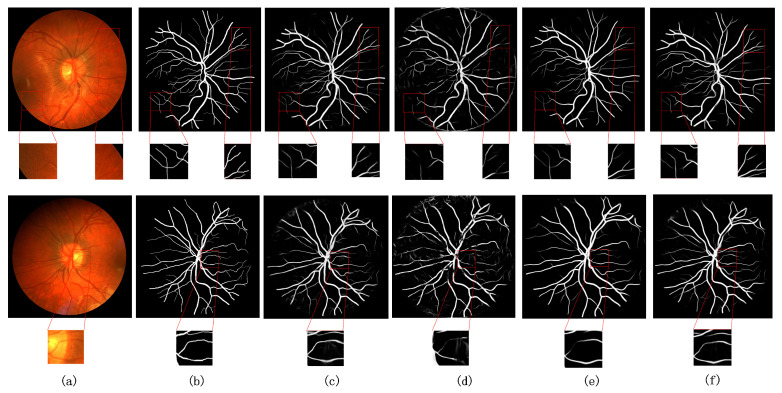
Comparison of visualization results with other methods on CHASE DB1 dataset (**a**) original image, (**b**) groundtruth image, (**c**) UNet++ [13], (**d**) CA-Net [9], (**e**) AG-UNet [52], (**f**) Ours.

**Figure 11 sensors-22-04592-f011:**
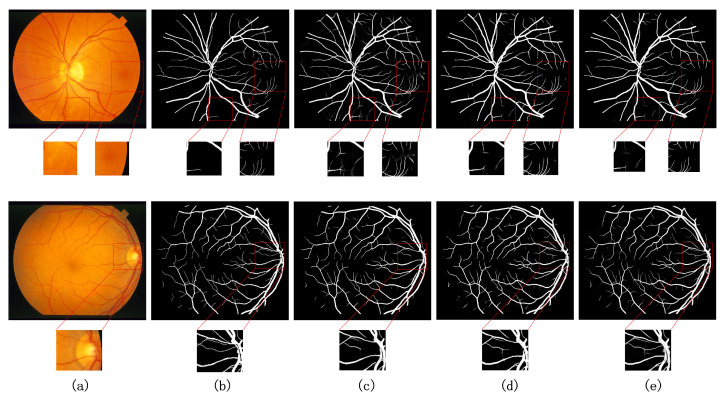
Comparison of visualization results with other methods on STARE dataset (**a**) original image, (**b**) groundtruth image, (**c**) UNet++ [13], (**d**) CA-Net [13], (**e**) Ours.

**Figure 12 sensors-22-04592-f012:**
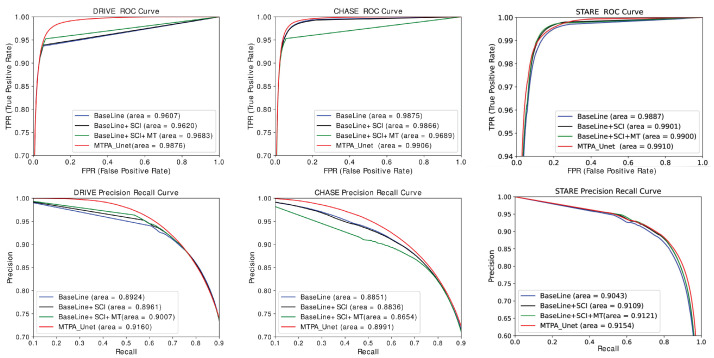
PR and ROC curves for each ablation structure.

**Table 1 sensors-22-04592-t001:** Ablation experiments on the DRIVE dataset.

Model	Accuracy	Specificity	Sensitivity	Dice	AUC_ROC
U-Net [10]	0.9531	0.9820	0.7537	0.8142	-
BaseLine	0.9705	**0.9865**	0.8060	0.8266	0.9728
BaseLine+SCI	0.9702	0.9841	0.8278	0.8289	0.9766
BaseLine+SCI+MT	0.9705	0.9848	0.8245	0.8300	0.9785
MTPA_Unet (w/o pre)	0.9702	0.9856	0.8116	0.8275	0.9875
MTPA_Unet	**0.9718**	0.9836	**0.8410**	**0.8318**	**0.9877**

The bold data in the table represent the maximum value achieved on each evaluation index.

**Table 2 sensors-22-04592-t002:** Ablation experiments on the CHASE DB1 dataset.

Model	Accuracy	Specificity	Sensitivity	Dice	AUC_ROC
U-Net [10]	0.9578	0.9701	0.8288	0.7783	-
BaseLine	0.9759	0.9857	0.8324	0.8136	0.9884
BaseLine+SCI	0.9761	**0.9861**	0.8296	0.8143	0.9878
BaseLine+SCI+MT	0.9757	0.9848	0.8434	0.8139	0.9861
MTPA_Unet (w/o pre)	0.9758	0.9857	0.8299	0.8122	0.9900
MTPA_Unet	**0.9762**	0.9858	**0.8437**	**0.8164**	**0.9905**

The bold data in the table represent the maximum value achieved on each evaluation index.

**Table 3 sensors-22-04592-t003:** Test results on STARE dataset using the leave-one-out method.

Image	Accuracy	Specificity	Sensitivity	Dice	AUC_ROC
0	0.9742	0.9825	0.8856	0.8483	0.9908
1	0.9753	0.9812	0.8916	0.8276	0.9892
2	0.9805	0.9859	0.8964	0.8468	0.9931
3	0.9694	0.9787	0.8525	0.8052	0.9866
4	0.9715	0.9824	0.8621	0.8455	0.9886
5	0.9778	0.9831	0.9061	0.8503	0.9921
6	0.9732	0.9744	0.9602	0.8521	0.9934
7	0.9775	0.9799	0.9484	0.8634	0.9941
8	0.9827	0.9872	0.9302	**0.8948**	**0.9958**
9	0.9755	0.9799	0.9257	0.8591	0.9923
10	0.9811	0.9876	0.8962	0.8713	0.9947
11	0.9812	0.9828	**0.9620**	0.8881	0.9954
12	0.9805	0.9864	0.9205	0.8939	0.9947
13	0.9803	0.9883	0.8997	0.8923	0.9946
14	0.9787	0.9871	0.8906	0.8783	0.9929
15	0.9684	0.9814	0.8542	0.8468	0.9879
16	0.9761	0.9864	0.8707	0.8671	0.9915
17	**0.9866**	**0.9935**	0.8571	0.8664	0.9926
18	0.9839	0.9903	0.8417	0.8191	0.9922
19	0.9721	0.9825	0.8254	0.7974	0.9883
Average	0.9773	0.9841	0.8938	0.8557	0.9920

The bold data in the table represent the maximum value achieved on each evaluation index.

**Table 4 sensors-22-04592-t004:** Ablation experiments on the STARE dataset.

Model	Accuracy	Specificity	Sensitivity	Dice	AUC_ROC
U-Net [10]	0.9690	**0.9842**	0.8270	0.8373	-
BaseLine	0.9747	0.9833	0.8701	0.8379	0.9901
BaseLine+SCI	0.9754	0.9834	0.8772	0.8425	0.9905
BaseLine+SCI+ MT	0.9766	0.9835	0.8802	0.8434	0.9908
MTPA_Unet	**0.9773**	0.9841	**0.8938**	**0.8557**	**0.9920**

The bold data in the table represent the maximum value achieved on each evaluation index.

**Table 5 sensors-22-04592-t005:** Comparison with other methods on the DRIVE dataset.

Type		Methods	Year	Acc	Sp	Se	Dice
Non-learning based methods		Azzopardi et al. [48]	2015	0.9442	0.9704	0.7655	-
		Miao et al. [49]	2015	0.9597	0.9748	0.7481	-
		Chen et al. [50]	2017	0.9390	0.9680	0.7358	-
		Shah et al. [6]	2019	0.9470	0.9724	0.7760	-
Deep learning based methods	Original image	U-Net [10]	2018	0.9531	0.9820	0.7537	0.8142
		Guo et al. [51]	2020	0.9691	0.9839	0.8149	0.8222
		AG-UNet [52]	2020	0.9558	0.9810	0.7854	0.8216
		FANet [53]	2021	0.8189	0.9826	-	0.8183
		Tong et al. [54]	2021	0.9684	**0.9870**	0.8117	0.8174
	Only pre-processing	SCS-Net [47]	2021	0.9697	0.9838	0.8289	0.8189
		PLRS-Net [33]	2022	0.9682	0.9817	0.8269	-
	Patch+ Pre-processing	RVSeg-Net [55]	2020	0.9681	0.9845	0.8107	-
		R2U-Net [46]	2018	0.9556	0.9813	0.7792	0.8171
		UNet++ [13]	2018	0.9710	0.9861	0.8120	0.8302
		DUNet [56]	2019	0.9566	0.9800	0.7963	0.8237
		CA-Net [9]	2020	0.9605	0.9788	0.7727	0.7733
		Huang et al. [57]	2021	0.9701	0.9849	0.8011	-
		Ours	2021	**0.9718**	0.9836	**0.8410**	**0.8318**

The bold data in the table represent the maximum value achieved on each evaluation index.

**Table 6 sensors-22-04592-t006:** Comparison with other methods on the STARE dataset.

Type		Methods	Year	Acc	Sp	Se	Dice
Deep learning based methods	Original image	U-Net [10]	2018	0.9690	0.9842	0.8270	0.8373
		Tong et al. [54]	2021	**0.9805**	**0.9927**	0.8072	0.8270
	Only pre-processing	SCS-Net [47]	2021	0.9736	0.9839	0.8207	-
		PLRS-Net [33]	2022	0.9715	0.9803	0.8635	-
	Patch+ Pre-processing	UNet++ [13]	2018	0.9753	0.9843	0.8646	0.8393
		R2U-Net [46]	2018	0.9712	0.9862	0.8298	0.8475
		Iter-Net [58]	2020	0.9782	0.9919	0.7715	0.8146
		CA-Net [9]	2020	0.9703	0.9705	0.8685	0.8397
		Ours	2022	0.9773	0.9841	**0.8938**	**0.8557**

The bold data in the table represent the maximum value achieved on each evaluation index.

**Table 7 sensors-22-04592-t007:** Comparison with other methods on the CHASE DB1 dataset.

Type		Methods	Year	Acc	Sp	Se	Dice
Non-learning based methods		Azzopardi et al. [48]	2015	0.9387	0.9587	0.7585	-
Deep learning based methods	Original image	U-Net [10]	2018	0.9578	0.9701	0.8288	0.7783
		AG-UNet [52]	2020	0.9752	0.9870	0.8110	0.8116
		FANet [53]	2021	0.7722	0.9830	-	0.8108
		Tong et al. [54]	2021	0.9739	**0.9868**	0.8340	0.7911
	Only pre-processing	SCS-Net [47]	2021	0.9744	0.9839	0.8365	-
		PLRS-Net [33]	2022	0.9731	0.9839	0.8301	-
	Patch+ Pre-processing	RVSeg-Net [55]	2020	0.9726	0.9836	0.8069	-
		UNet++ [13]	2018	0.9760	0.9810	0.8184	0.8139
		R2U-Net [46]	2018	0.9634	0.9820	0.7756	0.7928
		DUNet [56]	2019	0.9752	0.9610	0.8155	0.7883
		CA-Net [9]	2020	0.9645	0.9749	0.8120	0.7409
		Ours	2022	**0.9762**	0.9858	**0.8437**	**0.8164**

The bold data in the table represent the maximum value achieved on each evaluation index.

**Table 8 sensors-22-04592-t008:** Comparison of the number of parameters of each ablation structure network model.

Model	BaseBine	BaseBine + SCI	BaseBine + SCI + MT	MTPA_Unet
params	12.2 M	13.0 M	15.9 M	11.8 M

## Data Availability

Data availability statement. We use three publicly available retinal image datasets to evaluate the segmentation network proposed in this paper, namely, the DRIVE dataset, the CHASE DB1 dataset, and the STARE dataset. They can be downloaded from the URL http://www.isi.uu.nl/Research/Databases/DRIVE/ (accessed on 30 December 2021), https://blogs.kingston.ac.uk/retinal/chasedb1/ (accessed on 30 December 2021) and https://cecas.clemson.edu/~ahoover/stare/ (accessed on 31 December 2021).

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
