# Peer review of "MTPA_Unet: Multi-Scale Transformer-Position Attention Retinal Vessel Segmentation Network Joint Transformer and CNN"

_sensors, 2022, doi:10.3390/s22124592_

Round 1
Reviewer 1 Report
In this paper, the authors have proposed a Multi-scale Transformer-Position Attention Retinal Vessel Segmentation Network Joint Transformer and CNN. Overall paper is well written but requires the following improvements. 1) Authors need to tilt the introduction more toward Vessel segmentation 2) Figure 1 is unnecessary 3) Why DICE loss is used, the main attribute of DICE loss (class imbalance) is not discussed 4) Authors need to separate the methods (learning/non-learning based methods), (preprocess/ without preprocess), and patch-based/ full image-based methods in Result Tables because there will be a huge difference in overall processing time 5) please include the conventional U-Net in the Ablation study 6) Conclusion should reflect the claims in the Abstract without mentioning the Accuracy numbersAuthor Response
Please see the attachment.

Reviewer 2 Report
The authors presented a deep learning-based vessel segmentation method. I have following concerns:
(a) Authors should focus on retinal vessels mainly in the Introduction, and they should discuss a few diseases that are related to retinal vessel morphology
(b) I am confused with the contribution of the proposed method, by looking at the accuracy number it is fine but still the method is using preprocessing overhead, and also the scheme is based on a patch-based approach, Please elaborate the contribution of the paper (in comparison to the schemes that are using full image-based approaches and not using any preprocessing schemes for image enhancement) the points that are written in lines 103-119 are too long and looks apparent, please be concise and clear
(c) Unfortunately, I did not find the latest 2021-2022 papers in related work for vessel segmentation I would suggest following few
- Bridge-Net: Context-involved U-net with patch-based loss weight mapping for retinal blood vessel segmentation
- Retinal Vessel Segmentation with Skeletal Prior and Contrastive Loss
- Diabetic and Hypertensive Retinopathy Screening in Fundus Images Using Artificially Intelligent Shallow Architectures
- Detecting retinal vasculature as a key biomarker for deep Learning-based intelligent screening and analysis of diabetic and hypertensive retinopathy
- DF-Net: Deep fusion network for multi-source vessel segmentation
In related work discuss the merits and demerits of all the methods comparatively (in comparison to your methods)
(d) as per my knowledge, deep learning can avoid the requirement of preprocessing, you must test your method without processing (original images), for this, you can just experiment with the DRIVE dataset only (Provide the results in Table 1).
(e) Provide an architectural comparison (in a table) of your method with different versions of U-Net
(f) Do you use any post-processing? if yes, please explain
(g) How do you decide the optimal size of the patches (In Figure 2), other sizes will deteriorate the performance?
(h) what is the overall processing time of the proposed method (Patches+ image enhancement+ segmentation+ post-processing) for each dataset ?
(I) another question is how you measure the segmentation performance? did you do the patches for ground truth as well?? or did you resize the ground truth images? In medical images, it is not legal to change or resize the ground truth images at the testing end (because the images are from the medical experts and resizing it can remove the smaller information, which will not be reflected in the evaluation)
Round 2
Reviewer 1 Report
Most of my comments are addressed. I recommend acceptance of this article in its current form.
Reviewer 2 Report
The authors correctly responded to the comments. I vote for acceptance of this paper in current form.